# Review Article: Earth's ice imbalance

Thomas Slater[1], Isobel R. Lawrence[1], Inès N. Otosaka[1], Andrew Shepherd[1], Noel Gourmelen[2], Livia Jakob[3], Paul Tepes[2], Lin Gilbert[4]

[1]Centre for Polar Observation and Modelling, School of Earth and Environment, University of Leeds, LS2 9JT, UK
[2]School of GeoSciences, University of Edinburgh, Edinburgh, EH8 9XP, UK
[3]Earthwave Ltd, Edinburgh, EH9 3HJ, UK
[4]Mullard Space Science Laboratory, Department of Space & Climate Physics, University College London, WC1E 6BT, UK

*Correspondence to*: Thomas Slater (t.slater1@leeds.ac.uk)

**Abstract.** We combine satellite observations and numerical models to show that Earth lost 28 trillion tonnes of ice between 1994 and 2017. Arctic sea ice (7.6 trillion tonnes), Antarctic ice shelves (6.5 trillion tonnes), mountain glaciers (6.1 trillion tonnes), the Greenland ice sheet (3.8 trillion tonnes), the Antarctic ice sheet (2.5 trillion tonnes), and Southern Ocean sea ice (0.9 trillion tonnes) have all decreased in mass. Just over half (58 %) of the ice loss was from the northern hemisphere, and the remainder (42 %) was from the southern hemisphere. The rate of ice loss has risen by 57 % since the 1990s – from 0.8 to 1.2 trillion tonnes per year – owing to increased losses from mountain glaciers, Antarctica, Greenland, and from Antarctic ice shelves. During the same period, the loss of grounded ice from the Antarctic and Greenland ice sheets and mountain glaciers raised the global sea level by $34.6 \pm 3.1$ mm. The majority of all ice losses were driven by atmospheric melting (68 % from Arctic sea ice, mountain glaciers ice shelf calving and ice sheet surface mass balance), with the remaining losses (32 % from ice sheet discharge and ice shelf thinning) being driven by oceanic melting. Altogether, these elements of the cryosphere have taken up 3.2 % of the global energy imbalance.

## 1 Introduction

Fluctuations in Earth's ice cover have been driven by changes in the planetary radiative forcing (Vaughan et al., 2013), affecting global sea-level (The IMBIE Team, 2018, 2020; Zemp et al., 2019), oceanic conditions (Rahmstorf et al., 2015), atmospheric circulation (Francis and Vavrus, 2012; Vellinga and Wood, 2002) and freshwater resources (Huss and Hock, 2018; Immerzeel et al., 2020). Earth's cryosphere is created as meteoric ice in Antarctica, Greenland, and in mountain glaciers, and as frozen sea water in the Arctic and Southern oceans (Fig. 1). The polar ice sheets store more than 99 % (30 million km$^3$) of Earth's freshwater ice on land (Fretwell et al., 2013; Morlighem et al., 2017), and even modest losses raise the global sea level (The IMBIE Team, 2018, 2020), increase coastal flooding (Vitousek et al., 2017) and disturb oceanic currents (Golledge et al., 2019). To-date, these losses have tracked the upper range of climate warming scenarios forecast by the Intergovernmental Panel on Climate Change, which predict an ice sheet sea level contribution of up to 42 cm by 2100 (Slater et al., 2020). Ice sheet mass balance is the net balance between mass losses associated with ice flow, melting at the

ice-ocean interface, subglacial melt and the surface mass balance (the net difference between precipitation, sublimation, evaporation, wind erosion and meltwater runoff). Ice shelves are a major source of ocean fresh water (Jacobs et al., 1992), impart resistive forces on grounded ice upstream (buttressing), which would speed up in its absence (Weertman, 1974), and have been a persistent element of the climate system throughout the Holocene period (Domack et al., 2005). There are over
300 documented ice shelves (the vast majority of Earth's inventory) around Antarctica (SCAR, 2020; Shepherd et al., 2018) containing an estimated 380 thousand $km^3$ of ice (Fretwell et al., 2013), and fluctuations in their volume occur as a result of changes in their extent (Cook and Vaughan, 2010) and thickness (Adusumilli et al., 2020). Although ice shelves are much smaller and sparsely distributed across the Arctic, ice shelves fringing the northern coast of Ellesmere Island in Canada (Mortimer et al., 2012) and the Russian Arctic islands (Willis et al., 2015) have collapsed in recent decades. Mountain
glacier ice moderates global sea-level and regional hydrology (Huss and Hock, 2018), impacting local communities who rely on it as a source of freshwater (Immerzeel et al., 2020). There are over 215 thousand glaciers worldwide (RGI Consortium, 2017) containing 160 thousand $km^3$ of ice (Farinotti et al., 2019), and their retreat has accounted for 21 % of global sea-level rise between 1993 and 2017 (WCRP Global Sea Level Budget Group, 2018). Typically 15 to 25 million $km^2$ of the global ocean surface is covered in sea ice at any one time of year, though its thickness and extent vary seasonally and due to long-
term changes in Earth's climate (Maksym, 2019). Sea ice plays a key role in the freshwater and energy budgets of the polar regions and impacts the marine ecosystem (Stroeve and Notz, 2018), as well as regulating the absorption of solar radiation in summer (Pistone et al., 2014). Furthermore, sea ice loss could influence oceanic and atmospheric circulation and affect weather patterns in the mid-latitudes (Maksym, 2019; Vihma, 2014).

Although sparse in situ records of glacier mass balance date back to the 1890's (Zemp et al., 2015), substantial records of
change for other components of the cryosphere did not begin until the advent of satellite observations in the 1970's. Ice shelf extent has been recorded episodically in satellite imagery since the 1940's (Cook and Vaughan, 2010), sea ice extent has been monitored by satellites since the late 1970's (Cavalieri et al., 1999), and ice sheet, ice shelf, sea ice, and glacier thickness changes have been recorded systematically in satellite altimetry since the 1990s (Gardner et al., 2013; Laxon et al., 2013; Shepherd et al., 2010; The IMBIE Team, 2018, 2020). Here, we combine satellite observations of changing ice sheet,
ice shelf, glacier, and Arctic sea ice mass, with in situ and model-based estimates of glacier and Southern Ocean sea ice mass, to quantify trends in Earth's meteoric and oceanic ice. We do not include elements of the cryosphere that are not ice (i.e. snow on land and permafrost), or where knowledge of their global extent and change is limited (river and lake ice). However, these elements of the cryosphere have also experienced considerable change over recent decades: for example, it is estimated that the quantity of snow on land has decreased by $49 \pm 49$ gigatonnes per decade in the northern hemisphere since
1980 (Pulliainen et al., 2020); that permafrost (perennially frozen ground) has warmed globally by $0.29 \pm 0.12$ °C during the past decade (Biskaborn et al., 2019); and that the duration of river and lake ice cover has shortened by 12 days per century in the northern hemisphere over the last 200 years (Magnuson et al., 2000).

## 2 Mountain Glaciers

We combined eight estimates of mass change from an extrapolation of local glaciological and geodetic measurements (Zemp et al., 2019, 2020), satellite gravimetry (Wouters et al., 2019), satellite swath altimetry (Foresta et al., 2016; Jakob et al., 2020; Tepes et al., 2020) satellite synthetic differential aperture radar interferometry (DInSAR) (Braun et al., 2019), and satellite optical stereo images (Dussaillant et al., 2019; Shean et al., 2020), to produce a reconciled estimate of global glacier mass changes between 1962 and 2019 and over 19 glacier regions defined in the Randolph Glacier Inventory (RGI Consortium, 2017) (Fig. 2). Satellite gravimetry directly measures glacier mass change from fluctuations in Earth's gravitational field at monthly intervals, and as a result does not require knowledge of the density of the material lost or gained (Wouters et al., 2019). However, satellite gravimetry provides measurements at a spatial resolution on the order of hundreds of kilometres, which limits the interpretation of the spatial distribution of ice loss within individual glaciers. Satellite swath altimetry, DInSAR and optical stereo imagery all measure surface elevation change, which is converted to mass by assuming a fixed density of ice with an associated uncertainty of 60 kg/m$^3$ (Huss, 2013). Satellite swath altimetry uses the swath interferometric mode of CryoSat-2 which provides a dense grid of repeated elevation measurements (Foresta et al., 2016). CryoSat-2 swath altimetry provides up to two orders of magnitude more data than conventional altimetry processing, and homogeneous spatial coverage necessary to derive mass changes over relatively small glaciers with highly variable topography (Gourmelen et al., 2018; Jakob et al., 2020). The included DInSAR estimate measures surface elevation changes by differencing digital elevation models (DEMs) generated from the SRTM and TanDEM-X synthetic aperture radar missions (Braun et al., 2019). It is important to note that, for both satellite radar altimetry and DInSAR, the radar signal can penetrate beyond the glacier surface into snow and firn (Braun et al., 2019; Jakob et al., 2020); the impact of radar penetration on elevation measurements is difficult to quantify as it depends on spatiotemporal variations in snow and firn characteristics, and is an area of ongoing research. We also include estimates of glacier mass balance derived from satellite optical stereo imagery, which generates time series of high resolution DEMs from ASTER, Worldview-1/-2/-3 and GeoEye-1 satellite imagery (Dussaillant et al., 2019; Shean et al., 2020). In glacier regions where these estimates are available (High Mountain Asia, Southern Andes), they offer almost complete coverage of glaciated areas at high (metre-scale) resolution which can resolve changes within individual glaciers. However, optical imagery is weather-dependent and cloud cover can limit coverage in glacier regions. For each region we aggregated annual mass change rates determined from the techniques available: each region includes between 2 and 4 estimates except for glaciers peripheral to Antarctica and Greenland, where only estimates derived from the extrapolation of in-situ and geodetic data are available. For studies in which time-varying mass change rates are not available, we assume the mass change rate to be linear over the period considered and scale the uncertainty by the square root of the number of years. We computed the cumulative mass change as the integral of the aggregated mass change rates and accumulate the associated uncertainty over time as the root sum square of the annual errors. We summed the regional estimates to derive the global glacier mass change and the overall uncertainty as the root-mean square of the regional errors.

We assessed the consistency of the in situ and satellite gravimetry, altimetry and stereo imagery estimates between 2010 and 2015 in 7 regions (Arctic Canada North and South, Russian Arctic, Iceland, Svalbard and Jan Mayen, High Mountain Asia and Southern Andes) where measurements from all techniques overlap (Fig. 2). We record the largest difference (26 Gt yr$^{-1}$) and standard deviation (14 Gt yr$^{-1}$) between mass balance estimates in Arctic Canada North – the largest region included in our inter-comparison. The standard deviations of the mass change estimates are 9 Gt yr$^{-1}$, 8 Gt yr$^{-1}$, 6 Gt yr$^{-1}$, 5 Gt yr$^{-1}$, 2 Gt yr$^{-1}$ and 2 Gt yr$^{-1}$ for the Southern Andes, Russian Arctic, High Mountain Asia, Arctic Canada South, Iceland and Svalbard and Jan Mayen regions, respectively. Based on our reconciled estimate, glaciers have collectively lost -9,975 ± 1,667 Gt of ice between 1962 and 2019, raising the global mean sea-level by 27.7 ± 4.6 mm during this period. Glaciers peripheral to Greenland and in Alaska and the Southern Andes have experienced the largest losses (Fig. 2) – 5,694 ± 635 Gt between 1962 and 2019 – and account for more than half (57 %) of the global glacier mass loss over this period. Globally, the rate of glacier mass loss has increased from -120 ± 70 Gt yr$^{-1}$ in the 1970s to -327 ± 65 Gt yr$^{-1}$ between 2010 and 2019, peaking at -506 ± 192 Gt yr$^{-1}$ in 2018. Glacier mass loss is linked to increasing air temperatures; approximately 70 % of the global glacier mass loss has been attributed to anthropogenic forcing, and the remainder is due to natural climate variability (Marzeion et al., 2014).

## 3 Ice sheets

Ice sheets lose mass when ice discharge and melting at the surface and ice-ocean interface combined exceed snowfall. We use estimates of ice sheet mass balance and their uncertainty derived from an ensemble of satellite altimetry, satellite gravimetry and input-output datasets which span the period 1992-2018. For the Antarctic (24 datasets) (The IMBIE Team, 2018) and Greenland (26 datasets) (The IMBIE Team, 2020) ice sheets, independently derived estimates of mass change from the three satellite geodetic techniques were combined into a single estimate of ice sheet mass balance. Estimates of ice sheet mass balance derived from these methods at the continental scale are similar and can be collated to reduce uncertainty (The IMBIE Team, 2018, 2020): satellite altimetry directly measures changes in ice sheet height (Otosaka et al., 2019; Sandberg Sørensen et al., 2018) converted into mass by assigning a specific density to the volume change (Shepherd et al., 2019) or by explicitly accounting for snowfall fluctuations through firn modelling (Sørensen et al., 2011). Satellite gravimetry measures temporal variations in Earth's gravity field using spherical harmonic solutions (Velicogna et al., 2020) or through local mass concentration analysis (Luthcke et al., 2006). The input-output method removes ice discharge into the oceans (output), estimated from satellite observations of ice velocity and estimates of ice thickness, from the net snow accumulation (input) (Mouginot et al., 2019; Rignot et al., 2019) determined from regional climate modelling (Noël et al., 2018; van Wessem et al., 2018).

These satellite surveys (e.g. Fig. 1) show the Antarctic ice sheet lost 2,603 ± 563 Gt of ice between 1992 and 2017, and the Greenland ice sheet lost 3,902 ± 342 Gt of ice between 1992 and 2018. Since 2012, the rate of ice loss from Antarctica has

tripled when compared to the previous two decades, owing to widespread glacier speedup (Mouginot et al., 2014) and thinning (Shepherd et al., 2019) in the Amundsen and Bellingshausen Sea sectors in response to the circulation of warm water under the region's ice shelves (Jacobs et al., 2011). Ice shelf collapse (Cook and Vaughan, 2010) (Fig. 3) and thinning

at the Antarctic Peninsula has triggered speedup of glaciers upstream (Hogg et al., 2017) as a consequence of reduced ice shelf buttressing. Unlike in Antarctica, where almost all of the ice loss is associated with ice dynamical imbalance, just over half of Greenland's mass loss during this period arose due to increases in meltwater runoff (Enderlin et al., 2014) enhanced by atmospheric circulation during several warm summers (Bevis et al., 2019). The remaining ice loss was due to increased glacier discharge, primarily at Jakobshavn Isbræ (Holland et al., 2008) and at outlet glaciers in the southeast (Howat et al.,

2008) and northwest (Moon et al., 2012). Both ice dynamic and surface processes in Greenland have led to widespread thinning at the ice sheet margins and within individual glacier catchments (McMillan et al., 2016) (Fig. 1). Altogether, ice losses from Antarctica and Greenland have caused global sea levels to rise by 17.8 ± 1.8 mm between 1992 and 2017 (The IMBIE Team, 2018, 2020).

## 4 Antarctic ice shelves

To compute trends in the volume of Antarctic ice shelves associated with changes in their extent, we combined satellite-based records of their thickness (Fretwell et al., 2013) and area change (Cook and Vaughan, 2010) over time, adjusted for changes in thickness where they have been recorded (Adusumilli et al., 2020). We restrict this calculation to ice shelves at the Antarctic Peninsula, where a record of progressive retreat has been well-established (Fig. 3). Although area changes have been mapped since the late 1940's, comprehensive estimates of their thickness only began in the early 1990's. To estimate

the thickness of icebergs calved prior to this period, we combined in situ, airborne, and satellite-derived measurements of ice thickness recorded prior to when the ice shelf calving took place (Fig. 3). Uncertainties in volume change associated with ice shelf retreat were computed as the product of errors in ice thickness, determined from the variance of the thickness data, and extent, determined from the precision of the satellite imagery (Cook and Vaughan, 2010). We then used satellite altimetry to determine the volume changes of Antarctic ice shelves owing to changes in their thickness, and their associated uncertainty.

For this calculation, we use time series of ice thickness change and their estimated uncertainty derived by Adusumilli et al. (2020) from ERS-1, ERS-2, Envisat and CryoSat-2 satellite radar altimetry between 1994 and 2020, following the method of Paolo et al. (2015). Adusumilli et al. (2020) applied the following processing steps: (i) ice shelf surface elevation was computed by adjusting the altimeter range measurements for changes in ocean surface height, including contributions due to the geoid, mean dynamic topography, ocean tide, ocean load tide, atmospheric pressure, and sea-level rise; (ii) time series of

ice shelf elevation change were produced by grouping the elevation measurements within regularly spaced 10 km grid cells, applying a space-time polynomial fit to data from each mission; (iii) time series of ice shelf thickness change were calculated by adjusting the elevation change for fluctuations in firn air content and using a hydrostatic buoyancy relationship, assuming values of 917 and 1,028 kg/m$^3$ for the densities of ice and ocean water, respectively; and (iv) time series of ice shelf volume

change were computed from the thickness changes and using the minimum (fixed) area for each ice shelf. Full details of the methods used in this calculation can be found in Paolo et al. (2015). The total change in ice shelf volume is computed as the sum of changes due to thinning and retreat, and the uncertainty is estimated as the root sum square of the respective uncertainties.

Antarctic ice shelves have lost $8,667 \pm 1240$ Gt of their mass between 1994 and 2020, 54 % of which has been due to reductions in their extent and the remainder due to changes in their thickness. Although episodic iceberg calving is part of the natural cycle of ice mass transport through the continent, there has been a 39,717 km$^2$ loss of ice shelf area at the Antarctic Peninsula (e.g. (Cook and Vaughan, 2010)), where air temperatures have risen several times faster than the global trend (Vaughan et al., 2003). Warmer air leads to increased surface melting, which can promote iceberg calving through hydraulic fracture of crevasses (Scambos et al., 2013). At the same time, ocean-driven melting has caused some ice shelves to thin at their base, particularly in the Amundsen and Bellingshausen Seas (Paolo et al., 2015; Shepherd et al., 2010) where warm circumpolar deep water is present (Jacobs et al., 1996), but also at the Antarctic Peninsula (Shepherd et al., 2003). Ice shelf thinning can promote instability by weakening their lateral margins (Vieli et al., 2007). Both processes – calving front retreat and basal melting – have triggered speedup of inland ice (Rignot et al., 2004; Scambos et al., 2004; Shepherd et al., 2004) due to the associated reduction in buttressing (Joughin et al., 2012), leading to global sea-level rise (The IMBIE Team, 2018) even though ice shelves themselves are not a direct source of ocean mass. The ice shelf losses combined amount to 3 % of their present volume, while those in the Amundsen and Bellingshausen Seas are now 10 to 18 % thinner (Paolo et al., 2015) and those at the Antarctic Peninsula are 18 % smaller in extent (Cook and Vaughan, 2010).

**5 Sea ice**

We estimated trends in the mass of Arctic sea ice using a combination of sea-ice ocean modelling and satellite measurements of thickness change: between 1980 and 2011 we used the Pan-Arctic Ice-Ocean Modelling and Assimilation System (PIOMAS), a coupled sea ice-ocean model forced with atmospheric reanalyses (Zhang and Rothrock, 2003); from 2011, we used CryoSat-2 satellite radar altimetry measurements of sea ice volume (Tilling et al., 2018). We converted PIOMAS volume estimates to mass assuming a fixed density of 917 kg/m$^3$: this is the density used in the PIOMAS model to attribute a volume to the simulated sea ice growth (Schweiger, personal comm), therefore it is appropriate to convert PIOMAS volume estimates back to mass using this same density, as opposed to one that varies according to season or ice type. We divided CryoSat-2 monthly volume estimates into regions of multi-year and first-year ice and multiplied by densities of 882 kg/m$^3$ and 916.7 kg/m$^3$, respectively, to convert to mass (Tilling et al., 2018). The presence of melt ponds on the Arctic sea ice surface from May to September make it difficult to discriminate between radar returns from leads and sea ice floes, preventing the retrieval of summer sea ice thickness and volume from radar altimetry (Tilling et al., 2018). As a result, we computed the winter-mean (October to April) mass trend across the Arctic for both CryoSat-2 and PIOMAS estimates to

maintain consistency: the difference between winter (October-April) and annual (January-December) PIOMAS mass trends during 1980-2011 is 19 Gt yr$^{-1}$ (6 %) smaller when compared to the magnitude of the overall 12-month trend (-324 Gt yr$^{-1}$). Since the annual trend is slightly larger, we consider our winter-average mass trend to be a conservative estimate of the actual Arctic sea ice mass loss. In the absence of an available satellite-derived Antarctic sea ice volume product, we used the Global Ice-Ocean Modelling and Assimilation System (GIOMAS) (Zhang and Rothrock, 2003), the global equivalent to PIOMAS, to estimate the trend. We gridded GIOMAS sea ice thickness data onto 0.2 x 0.5 degree grids, multiplied by cell area to retrieve total volume and used a density of 917 kg/m$^3$ to convert to mass (as in PIOMAS, this is the density used to attribute a volume to the simulated sea ice growth in GIOMAS (Zhang, personal comm)). Antarctic sea ice trends were computed as annual averages between January and December. The uncertainties on PIOMAS volume for October and March are 1,350 and 2,250 km$^3$, respectively, estimated in Schweiger et al. (2011) using a range of methods, including comparison to in situ data and model sensitivity analyses, We take the average of these (1,800 km$^3$) as the uncertainty for all months and, in the absence of a formal error budget, we assign the same uncertainty to monthly GIOMAS estimates. We convert this monthly volume error of 1,800 km$^3$ to a mass error using the fixed PIOMAS/GIOMAS density of 917 kg/m$^3$. We estimated the uncertainty on monthly Arctic sea ice volume and mass from CryoSat-2 as a percentage uncertainty, which varies from 14.5 % volume in October to 13 % volume in April (Tilling et al., 2018). The uncertainty on the winter-average (Arctic) and annual-average (Antarctic) mass was propagated from the monthly uncertainties.  Finally, we estimated the uncertainty associated with a rate of mass change over a given time period by dividing the total error by the number of years.

Between the winters of 1980 (October 1979 to April 1980) and 2019 (October 2018 to April 2019), Arctic sea ice mass reduced by 230 ± 27 Gt yr$^{-1}$, predominantly due to a decline in the lateral extent of the ice cover (Fig. 1), which accounts for 93 % of the variance in volume over the entire PIOMAS record. The entire summer ice pack has thinned, largely attributable to the loss of the oldest and thickest ice, and sea ice cover has receded in the Beaufort, Chukchi and East Siberian seas (Stroeve and Notz, 2018). Arctic sea ice loss has been attributed to atmospheric warming driven by anthropogenic $CO_2$ emissions (Meredith et al., 2019; Stroeve and Notz, 2018), which has been enhanced in the Arctic when compared to the mid-latitudes likely due to sea ice loss itself (Dai et al., 2019; Screen and Simmonds, 2010).  Between 1980 and 2019, GIOMAS volume estimates, which incorporate observations of sea ice extent, show an increase in Antarctic sea ice of +43 ± 17 Gt yr$^{-1}$. No consensus has been reached on whether trends in Antarctic sea ice cover are anthropogenically driven, for example via the depletion of the Ozone layer (Ferreira et al., 2015), or the result of natural climate variability (Meehl et al., 2016; Zhang et al., 2019). Given the vastness of the continent it surrounds, regional analyses of Southern Ocean sea ice are essential to understand the processes driving it. The overall trend is a combination of sea ice thickening in the Weddell Sea and thinning in the Amundsen Sea (Fig. 1), accompanied by increases and reductions of the extent in each region, respectively (Parkinson, 2019). In general, global climate models predict a shrinking southern ice cap in response to climate change; projections from the latest coupled climate models suggest that Antarctic sea ice will decline during the 21[st] century (Roach et al., 2020).

## 6 Earth's ice imbalance

To determine the global ice imbalance, we summed the mass change of each ice component computed at annual intervals and estimated the combined uncertainty as the root sum square of the individual uncertainty estimates. Between 1994 and 2017, the Earth lost 27.5 ± 2.1 Tt of ice (Fig. 4) – at an average rate of 1.2 ± 0.1 Tt per year (Table 1). Ice losses have been larger in the northern hemisphere, primarily owing to declining Arctic sea ice (-7559 ± 1021 Gt) followed by glacier retreat (-5,148 ± 564 Gt) and Greenland ice sheet melt (-3,821 ± 323 Gt). Ice in the southern hemisphere from the ice shelves (-6,543 ± 1221 Gt), the Antarctic ice sheet (-2,545 ± 554 Gt), glaciers (-965 ± 729 Gt), and sea ice in the Southern Ocean (-924 ± 674 Gt) has been lost at a total rate of -477 ± 146 Gt yr$^{-1}$ – 34 % slower than in the northern hemisphere (-719 ± 207 Gt yr$^{-1}$). Earth's ice can be categorised into its floating and on-land components; grounded ice loss from ice sheets and glaciers raises the global sea-level (The IMBIE Team, 2018, 2020; Zemp et al., 2019), influences oceanic circulation through freshwater input (Rahmstorf et al., 2015) and glacier retreat impacts local communities who rely on glaciers as a freshwater resource (Immerzeel et al., 2020). Grounded ice losses have raised the global mean sea-level by 24.9 ± 1.8 mm and 9.7 ± 2.5 mm in the northern and southern hemispheres respectively, totalling 34.6 ± 3.1 mm over the 24-year period. Although the loss of floating sea ice and ice shelves does not contribute to global sea-level rise, sea ice decline increases habitat loss (Rode et al., 2014), coastal erosion (Overeem et al., 2011) and ocean circulation (Armitage et al., 2020), and may affect mid-latitude weather and climate (Blackport et al., 2019; Overland et al., 2016).

There is now widespread evidence that climate change has caused reductions in Earth's ice. On average, the planetary surface temperature has risen by 0.85 °C since 1880, and this signal has been amplified in the polar regions (Hartmann et al., 2013). Although this warming has led to higher snowfall in winter, it has also driven larger increases in summertime surface melting (Huss and Hock, 2018). The global oceans have warmed too (Hartmann et al., 2013), with significant impacts on tidewater glaciers (Hogg et al., 2017; Holland et al., 2008), on floating ice shelves (Shepherd et al., 2010), and on the ice streams which have relied on their buttressing (Rignot et al., 2004). Atmospheric warming – anthropogenic or otherwise – is responsible for the recent and long-term reductions in mountain glacier ice (Marzeion et al., 2014), and ocean-driven melting of outlet glaciers has caused the vast majority of the observed ice losses from Antarctica (The IMBIE Team, 2018). Elsewhere, the picture is more complicated. In Greenland, for example, roughly half of all ice losses are associated with trends in surface mass balance, and the remainder is due to accelerated ice flow triggered by ocean melting at glacier termini (The IMBIE Team, 2020). Although the retreat and collapse of ice shelves at the Antarctic Peninsula has occurred in tandem with a rapid regional atmospheric warming (Vaughan et al., 2003), warm circumpolar deep water has melted the base of ice shelves in the Amundsen and Bellingshausen Seas (Jacobs et al., 2011) and this now amounts to over half of their net loss. While the progressive retreat of Arctic sea ice has been driven by radiative forcing, this has been mediated in part by the increasing presence of open water (Perovich and Richter-Menge, 2009), and broader changes in oceanic conditions are expected to play an increasingly important role (Carmack et al., 2016). Finally, although the extent of Southern Ocean sea ice has shown little overall change, there have been considerable regional variations owing to changes in both atmospheric

and oceanic forcing (Hobbs et al., 2016). Attributing Arctic sea ice decline and ice shelf calving to increased radiative forcing, approximately 68 % of the recent global ice imbalance is due to atmospheric warming, and the remainder is due to ocean-driven melting. We determine the energy required to melt the total ice loss as:

$$E = M(L + c_p \Delta T) \, , \qquad\qquad (1)$$

where $M$ is the mass of ice, $\Delta T$ is the rise in temperature required (we assume an initial ice temperature of -20 ± 10 °C), $L$ is the latent heat of fusion for water (333 J g$^{-1}$), and $c_p$ is the specific heat capacity of water (2108 J Kg$^{-1}$ °C$^{-1}$). Although the initial temperature is poorly constrained, the fractional energy required for warming is a small (0.7 % °C$^{-1}$) percentage of the total energy imbalance. Altogether, the ice sheet, glacier, ice shelf and sea ice loss amounts to an 8.9 ± 0.9 x 10$^{21}$ J sink of energy, or 3.2 ± 0.3 % of the global imbalance over the same period (Schuckmann et al., 2020).

**7 Conclusions**

Even though Earth's cryosphere has absorbed only a small fraction of the global energy imbalance, it has lost a staggering 28 trillion tonnes of ice between 1994 and 2017. The loss of grounded ice during this period has caused sea-levels to rise by 34.6 ± 3.1 mm, and the loss of floating ice has caused reductions in the planetary albedo (Thackeray and Hall, 2019), reductions in the buttressing of grounded ice (Rignot et al., 2004), ocean freshening (Jacobs et al., 1996), and ocean cooling

(Bintanja et al., 2013). Our assessment is based primarily on observations; we use satellite measurements to determine Antarctic and Greenland ice sheet mass balance and to determine changes in the mass of Antarctic ice shelves associated with retreat and thinning, we use a combination of satellite observations and *in situ* measurements to determine changes in the mass of mountain glaciers, and we use a combination of numerical models and satellite observations to determine changes in the mass of sea ice. There is generally good agreement in mass trends derived from observations and models,

where both are available. Only our estimate of Southern Ocean sea ice mass imbalance depends on modelling alone (Zhang and Rothrock, 2003), though satellite observations of changes in its extent (Parkinson, 2019) and in situ observations of changes in its thickness (Worby et al., 2008) suggest that little change has occurred in Antarctic sea ice cover. The overall rate of ice loss has increased by 57 % over the past 24 years compared to the 1990s, and *in situ* measurements of changes in glacier mass (Zemp et al., 2019) and satellite records of ice shelf extent (Cook and Vaughan, 2010) which pre-date the

complete survey confirm this trend. Although a small fraction of mountain glacier losses are associated with retreat since the little ice age (Marzeion et al., 2014), there can be little doubt that the vast majority of Earth's ice loss is a direct consequence of climate warming.

**Data availability**

Mountain glacier mass change data from glaciological and geodetic observations are freely available at (https://doi.org/10.5281/zenodo.1492141). Elevation change fields from DInSAR are available via the World Data Center (https://doi.org/10.1594/PANGAEA.893612). Glacier digital elevation models and elevation change maps derived from satellite optical stereo imagery are available at (https://nsidc.org/data/highmountainasia) and (https://doi.pangaea.de/10.1594/PANGAEA.903618), respectively. Mass change data for the Antarctic and Greenland ice sheets are provided by the ice sheet mass balance intercomparison exercise (IMBIE) and are available at (http://imbie.org/data-downloads/). Changes in ice shelf extent can be downloaded from the Scientific Committee on Antarctic Research digital database (https://www.add.scar.org/). Changes in ice shelf thickness from Adusumilli et al. (2020) are freely available at (https://doi.org/10.6075/J04Q7SHT). PIOMAS/GIOMAS data are freely available from the University of Washington Polar Science Data Center (http://psc.apl.uw.edu/data/).

**Author contributions**

TS, IRL, INO and AS designed the study, performed the data analysis and wrote the manuscript. NG, LJ and PT prepared mountain glacier mass change estimates from CryoSat-2 satellite radar altimetry. LG prepared the ice sheet thickness change datasets from multi-mission satellite radar altimetry used in Fig. 1.

**Competing interests**

The authors declare that they have no conflict of interest.

**Acknowledgements**

We thank Axel Schweiger and Jinlun Zhang for their help with PIOMAS/GIOMAS data and Susheel Adusumilli for providing ice shelf thickness change data. This work was supported by the Natural Environment Research Council Centre for Polar Observation and Modelling (grant number cpom300001).

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

|  | 1980s | 1990s* | 2000s | 2010s** | 1994-2017 |
|---|---|---|---|---|---|
| Arctic Sea Ice | -156 ± 88 | -298 ± 88 | -360 ± 88 | -94 ±119 | -329 ± 44 |
| Antarctic Sea Ice | +196 ± 67 | -27 ± 67 | +71 ± 67 | -83 ± 75 | -40 ± 29 |
| Ice Shelves Calving | -140 ± 15 | -125 ± 25 | -176 ± 57 | -250± 68 | -155 ± 36 |
| Ice Shelves Thinning | - | -19 ± 52 | -233 ± 57 | -53 ± 71 | -129 ± 39 |
| *Total Floating Ice* | - | *-469 ± 125* | *-698 ± 137* | *-480 ± 172* | *-653 ± 75* |
| Antarctic | - | -55 ± 38 | -78 ± 37 | -206 ± 47 | -111 ± 24 |
| Greenland | - | -34 ± 24 | -166 ± 21 | -247 ± 23 | -166 ± 14 |
| Glaciers | -62 ± 66 | -206 ± 63 | -252 ± 60 | -327 ± 65 | -266 ± 41 |
| *Total Grounded Ice* | - | *-296 ± 77* | *- 495 ± 74* | *-779 ± 83* | *-543 ± 49* |
| **Total** | - | **-764 ± 147** | **-1193 ± 156** | **-1259 ± 191** | **-1196 ± 90** |

*1990s: the decade is not entirely surveyed but starts from 1994 for Ice Shelves Thinning, and from 1993 for Antarctica and from 1992 for Greenland

**2010s: the decade is not entirely surveyed but covers up to 2016 for the Antarctic ice sheet, up to 2017 for Greenland, and up to 2019 for sea ice, glaciers and ice shelf calving.


**Table 1  Average mass change rates (Gt yr$^{-1}$) of the different global ice components, total floating ice, total grounded ice and global total per decade and over the common period 1994-2017.**

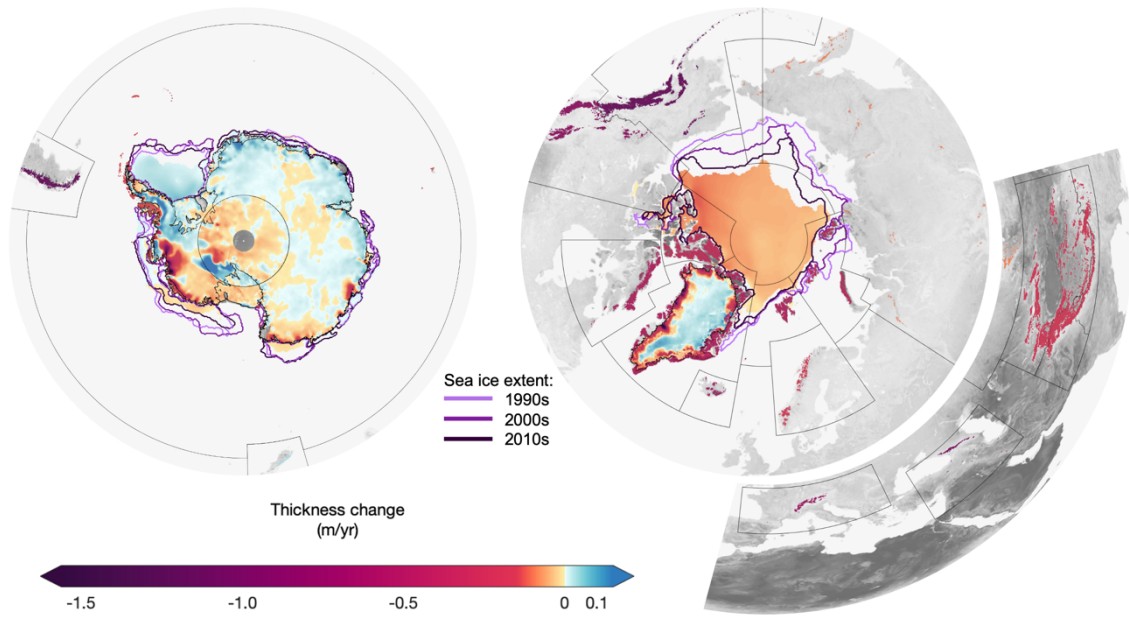


**Figure 1 Average rate of ice thickness change in the (left) southern and (right) northern hemispheres. Changes in Antarctic (1992-2017) and Greenland ice sheet (1992-2018) thickness were estimated using repeat satellite altimetry following the methods of (Shepherd et al., 2019). Sea ice thickness trends between 1990 and 2019 are determined from numerical sea ice and ocean modelling (Zhang and Rothrock, 2003), as well as the average minimum of sea ice extent in February (Antarctic) and September (Arctic) (purple lines) for each decade during the same period. Glacier thickness change between 1992 and 2018 for glacier regions defined in the Randolph Glacier Inventory (RGI Consortium, 2017) (black boundaries) are from mass change estimates (Braun et al., 2019; Foresta et al., 2016; Jakob et al., 2020; Tepes et al., 2020; Wouters et al., 2019; Zemp et al., 2019) which have been converted to a thickness change assuming an ice density of 850 kg/m3. The black circle at the south pole indicates the southern limit of the orbit of ERS and ENVISAT satellite altimeters, which were in operation between 1992 and 2010. The area between 81.5° and 88° S has been covered by CryoSat-2, which launched in 2010.**

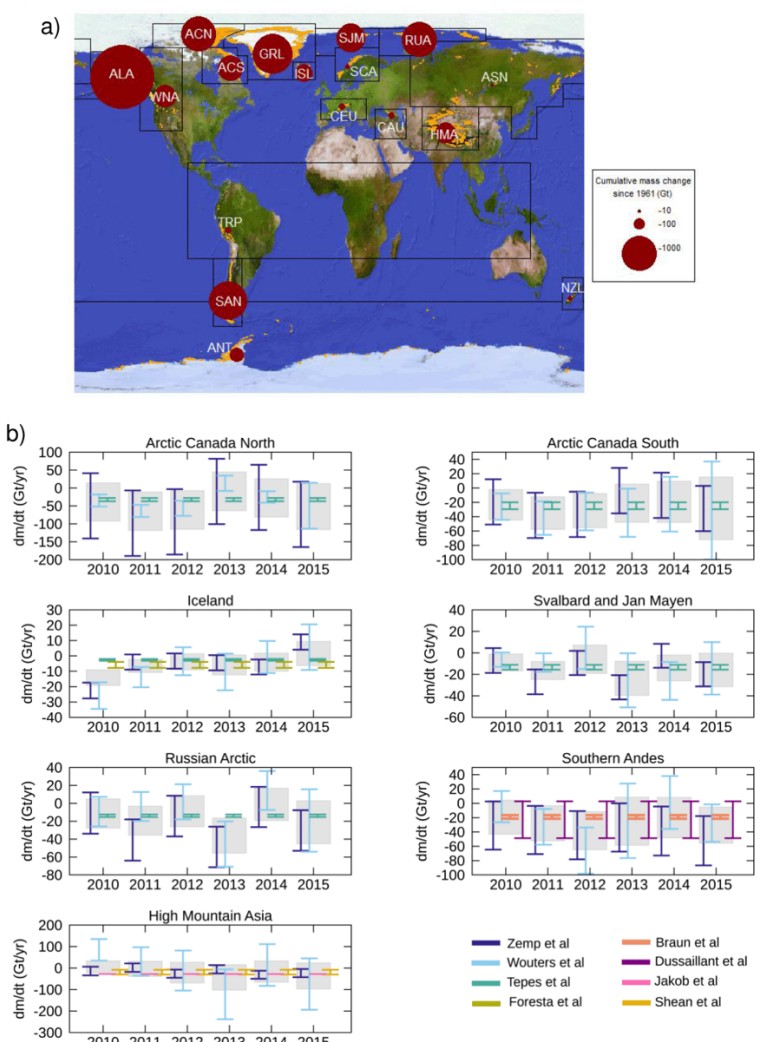

**Figure 2 (a) Cumulative mass change for glacier regions (Gt) between 1962 and 2019. Outlines of the glacier regions (RGI 6.0) are marked by black lines and glacierized areas are indicated in orange: ACN, Arctic Canada North (area 105,110 km²); ACS, Arctic Canada South (40,888 km²); ALA, Alaska (86,725 km²); ANT, Antarctic and Subantarctic (132,867 km²); CAU, Caucasus and Middle East (1,307 km²); CEU, Central Europe (2,092 km²); GRL, Greenland (89,717 km²); HMA, High Mountain Asia (97,606 km²); ISL, Iceland (11,059 km²); NZL, New Zealand (1,161 km²); RUA, Russian Arctic (51,591 km²); SAN, Southern Andes (29429 km²); SCA, Scandinavia (2,949 km²); SJM, Svalbard and Jan Mayen (33,958 km²); TRP, Low Latitudes (2,341 km²); WNA, Western Canada and USA (14,524 km²). (b) Glacier rate of mass change (Gt yr⁻¹) in regions where estimates from different techniques are available, including satellite altimetry (Foresta et al., 2016; Jakob et al., 2020; Tepes et al., 2020), extrapolation of in-situ glaciological and geodetic data (Zemp et al., 2019, 2020), satellite gravimetry (Wouters et al., 2019), satellite InSAR (Braun et al., 2019), and satellite stereo imagery (Dussaillant et al., 2019; Shean et al., 2020) over the period 2010-2015. The reconciled estimate (calculated as the average of the estimates available in a given region and year) is shown in grey.**

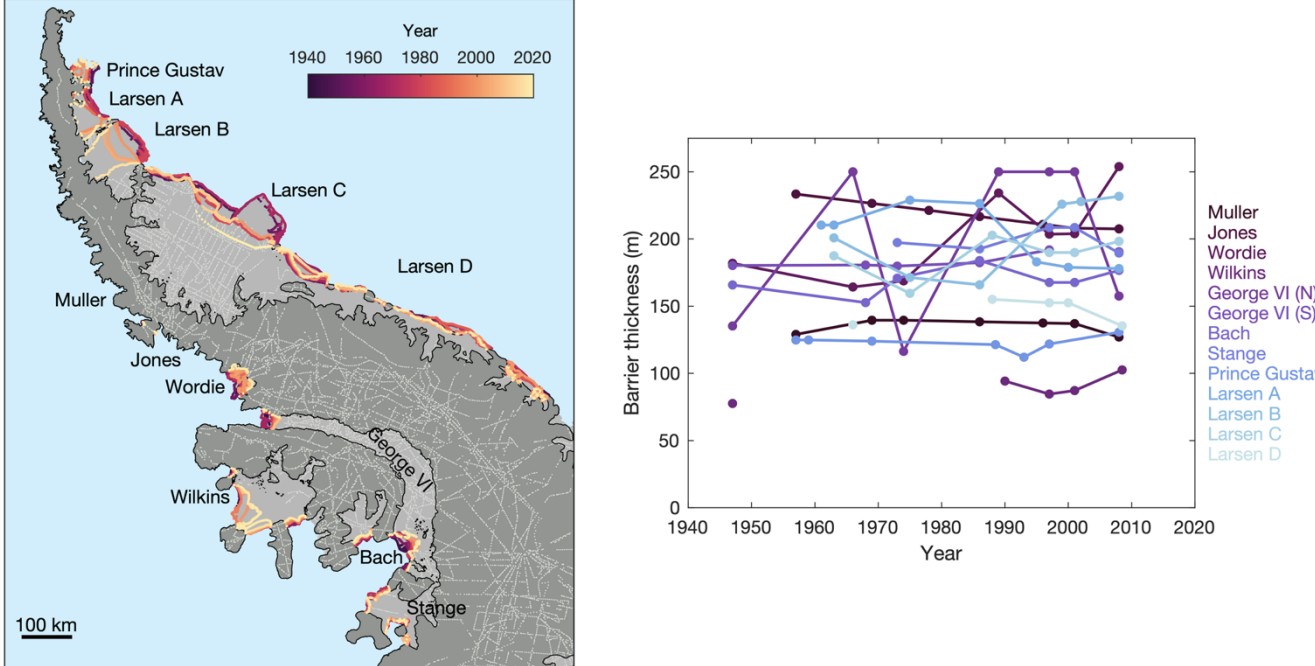

**Figure 3 Change in Antarctic ice shelf barrier position (left) and thickness (right) over time. Barrier positions are derived from episodic satellite imagery (Cook and Vaughan, 2010), and barrier thicknesses are derived from airborne ice penetrating radar (light grey lines) and satellite radar altimetry (Fretwell et al., 2013). Iceberg calving is calculated as the difference in area between successive barrier positions.**

620

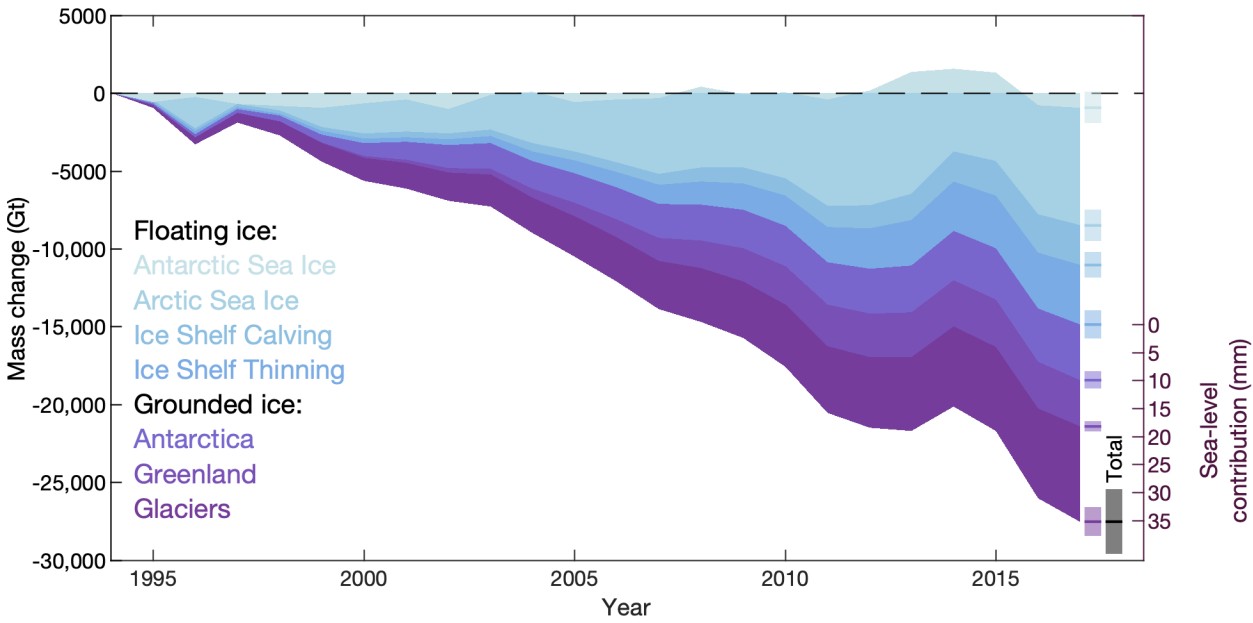

**Figure 4 Global ice mass change between 1994 and 2017 partitioned into the different floating (blues) and grounded (purples) components. Shaded bars indicate the cumulative mass change and estimated uncertainty for each individual ice component (blues, purples) and their sum (black). The equivalent sea-level contribution due to the loss of grounded ice from Antarctica, Greenland and mountain glaciers is shown in the y-axis on the right hand side.**