# Peer review of "Review Article: Earth's ice imbalance"

_The Cryosphere, 2020_

## Short Comment (SC1) · 23 Aug 2020

And the melt rate continues to rise. Exponentially?

---

## Referee Comment (RC1) · Anonymous Referee #1 · 8 Oct 2020

Review of 'Earth's ice imbalance' by Slater et al.,

General Comments: This manuscript uses a variety of satellite observation and numerical models in order to quantify mass change of the Earth's glaciers and ice caps, the ice sheets, sea ice and Antarctic ice shelves. Overall, the manuscript is clear, well written and clearly addresses the stated objectives. Generally, the methods utilized here have been well utilized in past studies and are appropriate for this work. Given this, I only have minor comments listed below for the authors to address and consider for a revised version of this manuscript. Specific items are provided below.

Specific Comments

L41: "...impacts on their marine ecosystem..." suggest change to "... impacts the marine ecosystem..."

[Figure]

L43: I think it would be worthwhile to note that although in situ records may have been obtained as early as the 1890s, that this would have been limited to only a very few locations.

L50-51: I would revise this last line. I think it is fine to note that trends in permafrost, terrestrial snow and lake/river ice are beyond the scope of this study. However, I would advise removing "which are small in comparison". This might be true, but that should not necessarily diminish the importance of the changes in these elements of the cryosphere.

L58: "within local mass concentration units" can you describe this more fully? Doesn't this just mean "within a region"? Remember this is a review article and the terminology/description needs to be understandable by a wide readership.

L60-63: Can you provide more detail here on CryoSat-2 and diff SAR InSAR methods? In particular, can you comment on whether variable penetration of the radar energy into glacier/ice surface impacts the uncertainty in the elevation differencing? Related to CryoSat-2, are there any issues in terms of using CryoSat-2 to measure changes in glaciated regions with significant topography? If these are not issues, please state why they are not. Related to CryoSat-2 data, can you mention the product level used? Was this an ESA distributed product or a custom derived product created by the authors? Related to DInSAR, can you mention the sensors utilized (broadly) to create the DEM products?

114-180: I know that this section is focused on Antarctic Ice Shelf changes, but you may also want to mention here (or elsewhere if more appropriate) recent changes to Arctic Ice Shelves as well. You do not necessarily need to quantify these changes but they should at least get a mention if this is a review paper of the changes of Earth's ice.

L170-175: Somewhere here you may want to quickly discuss where the major declines in Arctic sea ice occur spatially. And how does this match with other studies?

END OF REVIEW

---

## Referee Comment (RC2) · Anonymous Referee #2 · 27 Oct 2020

This manuscript (MS) gives a nice review of the ice imbalance on the Earth and a direct measure of global climate change, when it in the last part of the MS relates to the energy needed to melt the ice. The MS is generally well written and gives an overview of the methodology used in assessing each of the components. However, the level of references is kept to a minimum and mostly citing work directly related to numbers derived in this MS. Relying only on a limited number of studies makes the error propagation hard to assess. The title of the paper may also lead the reader to think it is a review of earth ice balance estimates, I would suggest the title to be more specific such as "Budget of the Earth meteoric ice masses". In the view of a budget-study, I only have minor comments to the manuscript.

L.16: Suggest removing "from"

L.19: It is not clear from the title nor abstract that you only consider meteoric ice. I

suggest adding "not accounting for changes in permafrost, . . .". The melting permafrost will also require energy from the "energy imbalance.", and therefore adding to the error estimate of the derived numbers in the MS.

L24: suggesting rephrasing: Meteoric ice is stored in the Earth's cryosphere land components; Ice sheets, Ice caps and glaciers, and its ocean components; ice shelves and sea ice (fig 1).

L.29: subglacial melt and evaporation are missing.

L31: as the manuscript is relevant for a broader audience, please use a few more words on the Weertman reference.

L32: are there 343 ice shelves or more than 340?

L35 remove "on"

L42: Missing a couple of lines about the change in Earth albedo with the loss of sea ice.

L50: The smaller components will add to the imbalance, please give numbers on the approximate magnitude of these components.

L53: "6"-> "six"

L60: The accumulation area is limited, but assuming fixed ice-density in the volume to mass conversion of the glaciers is an overestimate, this should be included in the uncertainty.

L64: please list the observational sources which are the only estimate available.

L67: How is the uncertainty propagated in time? This is a more general question as it is general hard to grasp from the MS how uncertainty is propagated, both for the individual estimates and the total imbalance. Please add information in the relevant sections of the manuscript.

Fig2: please change the colors so the studies can be separated.

L72: What is a close agreement? Please report the magnitude of 1 standard deviation, it is not readable from the figure. Maybe add a table (maybe in supp.).

L77: -10,130pm1713 Gt -> -10,130pm1,713 or -10130pm1713. This is a general comment and should follow the same convention throughout the MS. You also give Tt in some cases please consider the number of significant digits.

L96: Much of the uncertainty in the ice thickness is in the areas where it is modeled. "measured"-> "estimated".

L117: this statement with the reference to fig 3 is hard to follow as figure 3 only shows the peninsula. Please be specific as to which ice shelves are included in the study.

Fig 3 the color bar for time is not readable.

L120: Again, how is figure 3 informing on calving? Please add information in the caption to help the reader to better understand the figure.

L127: A polynomial-fit to elevation data is not trivial, which parameters are included in this, please elaborate on this equation and give references.

L130: the altimetry is adjusted for firn air, guess this is from RACMO, but not mentioned in the text. Additional information about the firn would be of high relevance.

L150: The reference guidelines https://www.the-cryosphere.net/submission.html#references states "Informal or so-called 'grey' literature may only be referred to if there is no alternative from the formal literature". Please give an argument for not having other references and therefore need to use "personal comm". This appears multiple places in the MS. Then I will leave it with the editor to see if this is justified.

The sea ice section more generally: For the density of sea-ice "glacier-ice" density is chosen for PIOMAS and GIOMAS, which is normally associated with first-year ice. Timco and Frederking (1996) report "in situ density of first-year sea ice range from

0.84 to 0.91 Mg m−3 for the ice above the waterline, and 0.90 to 0.94 Mg m−3 for the ice below the waterline." Please comment on the volume conversion and the effects on the uncertainty conversion. The neglect of multi-year ice densities in the model estimates (multi-year is included in Tilling's CS2 estimate) seems to overestimate the mass derived from the models. This should at least be included in the uncertainty.

L165: how is this uncertainty derived?

L186: The text states 1.2pm0.3 and the table states 1.2pm0.9, which is the right number?

Table1: As the periods are not the same it would be informative if the numbers could be given both as rates and totals in this table.

L220: Is the temperature of -20 used for all ice bodies? E.g. -20 seem cold for sea-ice with snow on top and water below.

L222: What is the uncertainty on this number? Figure 4 please add the cumulative uncertainty.

Reference: Timco and Frederking, A review of sea ice density Cold Reg. Sci. Technol., 24 (1996)

---

## Author Comment (AC1) · 15 Nov 2020

Thank you to the reviewers for their positive and constructive reviews of our manuscript. We have revised the manuscript in order to address the comments raised by each reviewer. In summary, the main changes are:

- We have revised the introduction to make clear why we do not consider snow on land, permafrost and river and lake ice (please see our responses to reviewer comments below for more details) and we have added example summaries of their changes over time.
- We have expanded upon details of the observational techniques used to measure glacier mass balance. At the request of Etienne Berthier (LEGOS), who got in touch with us privately, we have included two more estimates of glacier mass changes derived from optical stereo images to expand upon the amount of observational data used in our glacier estimate. These have had a minimal impact on our glacier (4 Gt/yr, 2%) and global ice loss (< 1%) estimates during the common period 1994-2017.
- Redesigns of Figures 2 and 3 for readability, and 4 to include cumulative errors.
- We have expanded on the computation of our uncertainty estimates across the manuscript.
- Several changes and additions to the text with improvements suggested in the reviewers' comments.

We have responded to the comments of each individual reviewer in full below. Reviewer comments are in *black italic*, and our responses are in blue. We have included a tracked changes version of the manuscript after our responses, so it is clear where changes have been made. We believe these changes have substantially improved the manuscript; thank you to both of the reviewers for their comments.

**Anonymous Referee #1**

*This manuscript uses a variety of satellite observation and numerical models in order to quantify mass change of the Earth's glaciers and ice caps, the ice sheets, sea ice and Antarctic ice shelves. Overall, the manuscript is clear, well written and clearly addresses the stated objectives. Generally, the methods utilized here have been well utilized in past studies and are appropriate for this work. Given this, I only have minor comments listed below for the authors to address and consider for a revised version of this manuscript. Specific items are provided below.*

Thank you for your review of our manuscript – we have addressed each of your specific comments below.

*L41 – "...impacts on their marine ecosystem..." suggest change to "...impacts the marine ecosystem..."*

Done.

*L43 – I think it would be worthwhile to note that although in situ records may have been obtained as early as the 1890s, that this would have been limited to only a very few locations.*

Thanks, we have amended the sentence to reflect this.

*L50-51 – I would revise this last line. I think it is fine to note that trends in permafrost, terrestrial snow and lake/river ice are beyond the scope of this study. However, I would advise removing "which are small in comparison". This might be true, but that should not necessarily diminish the importance of the changes in these elements of the cryosphere*

We do not intend to diminish the importance of these elements of the cryosphere, so we have revised this part of the text to (1) better explain why they are beyond the scope of this study and (2) provide more information to the reader on recent changes in these elements.

*L58 – "within local mass concentration units" can you describe this more fully? Doesn't this just mean "within a region"? Remember this is a review article and the terminology/description needs to be understandable by a wide readership."*

Mass concentration units are gravity field basis functions to which GRACE observations are fitted. We agree this is not accessible for a wide readership, so we have removed this to avoid confusion for the reader – providing a detailed technical background on the techniques used to derive glaciers mass changes falls outside the scope of this review paper, and adds a layer of complexity that is not necessary for a wider readership to appreciate the results presented. For the interested readers, additional information can be found in the studies we have cited.

*L60-63 – Can you provide more detail here on CryoSat-2 and diff SAR InSAR methods? In particular, can you comment on whether variable penetration of the radar energy into glacier/ice surface impacts the uncertainty in the elevation differencing? Related to CryoSat-2, are there any issues in terms of using CryoSat-2 to measure changes in glaciated regions with significant topography? If these are not issues, please state why they are not. Related to CryoSat-2 data, can you mention the product level used? Was this an ESA distributed product or a custom derived product created by the authors? Related to DInSAR, can you mention the sensors utilized (broadly) to create the DEM products?*

We have added more details on the swath altimetry and DInSAR methods. Also at the request of Etienne Berthier (LEGOS), who got in touch with us privately, we have included two more estimates of glacier mass changes derived from optical stereo images to expand upon the amount of observational data used in our glacier estimate: one for the Southern Andes and for the High Mountain Asia region. These have had a minimal impact on our glacier (4 Gt/yr, 2%) and global ice loss (< 1%) estimates during our common period.

*L114-180 – I know that this section is focused on Antarctic Ice Shelf changes, but you may also want to mention here (or elsewhere if more appropriate) recent changes to Arctic Ice Shelves as well. You do not necessarily need*

*to quantify these changes but they should at least get a mention if this is a review paper of the changes of Earth's ice*

Thanks for bringing this to our attention – as this section focuses on Antarctic ice shelf change we feel it is more appropriate to include this in the introduction, where we have added a sentence about recent changes in Arctic ice shelves.

*L170-175 Somewhere here you may want to quickly discuss where the major declines in Arctic sea ice occur spatially. And how does this match with other studies?*

Thanks – we have added this to the text.

**Anonymous Referee #2**

*This manuscript (MS) gives a nice review of the ice imbalance on the Earth and a direct measure of global climate change, when it in the last part of the MS relates to the energy needed to melt the ice. The MS is generally well written and gives an overview of the methodology used in assessing each of the components.*

Thank you for your review of our manuscript – please find responses to each of your general and specific comments below.

*However, the level of references is kept to a minimum and mostly citing work directly related to numbers derived in this MS. Relying only on a limited number of studies makes the error propagation hard to assess.*

We have expanded on the computation of our uncertainty estimates, where requested in your specific comments, in the hope that this is now more clear.

*The title of the paper may also lead the reader to think it is a review of earth ice balance estimates. I would suggest the title to be more specific such as "Budget of the Earth meteoric ice masses". In the view of a budget-study, I only have minor comments to the manuscript.*

Thank you for this suggestion, but we prefer our original title as it is an accurate and understandable description of the topic (Earth's ice).

- The term "meteoric" refers to ice of meteorological origin; it is not appropriate here because ice shelves and sea ice derive significant parts of their mass from the ocean as well ("oceanic").

- We feel that "ice imbalance" is clearer to a wider audience than "Budget of the …masses".

*L16 – Suggest removing "from"*

Done.

*L19 – It is not clear from the title nor abstract that you only consider meteoric ice.*
*I suggest adding "not accounting for changes in permafrost,…". The melting permafrost will also require energy*
*from the "energy imbalance.", and therefore adding to the error estimate of the derived numbers in the MS*

As we mentioned in response to your earlier comment, "meteoric" refers to ice of meteorological origin; ice shelves and sea ice derive significant parts of their mass from the ocean as well, so it is therefore not appropriate for us to say we only consider meteoric ice. The term "permafrost" is also not appropriate as it refers to frozen ground, not ice. We have, however, amended the introduction to clarify that we are considering both meteoric and oceanic ice.

We have also amended the introduction to explain that we do not consider elements of the cryosphere that are not ice (i.e. snow on land and permafrost) or river and lake ice, and we have added example summaries of their changes over time. We have also clarified in the abstract and main text that our assessment of the global energy imbalance pertains to the elements of the cryosphere we have considered.

*L24 – Suggesting rephrasing: Meteoric ice is stored in the Earth's cryosphere land com-ponents; Ice sheets, Ice caps and glaciers, and its ocean components; ice shelves and sea ice (fig 1)*

Thank you for this suggestion - however we would prefer to stick with our original wording as we feel the suggested sentence implies sea ice and ice shelves are solely meteoric in origin, which is not the case (they are partly oceanic).

*L29 – subglacial melt and evaporation are missing*

Thanks, we have added these to the sentence.

*L31 – As the manuscript is relevant for a broader audience, please use a few more words on the Weertman reference*

We have amended this clause to make this clearer to a broader audience.

*L32 – are there 343 ice shelves or more than 340?*

We have amended to read "over 300" (the database evolves and this was the value reported in Shepherd et al., 2018).

*L35 – remove "on"*

Done.

*L42 – Missing a couple of lines about the change in Earth albedo with the loss of sea ice*

Thanks, we have added this to the text.

*L50 – The smaller components will add to the imbalance, please give numbers on the approximate magnitude of these components*

This sentence was also flagged by reviewer 1: we have amended this sentence to explain that we do not consider elements of the cryosphere that are not ice (i.e. snow on land and permafrost) or river and lake ice, and we have added example summaries of their changes over time.

*L53 – "6"-> "six"*

Done.

*L60 – The accumulation area is limited, but assuming fixed ice-density in the volume to mass conversion of the glaciers is an overestimate, this should be included in the uncertainty.*

Here we are reporting a published mass balance estimate and its associated uncertainty as derived by the authors of the study – we feel it is beyond the scope of a review paper to amend single mass balance estimates and uncertainties which have already been published in the literature. However, we have added some text here to provide more information on the different techniques and their limitations to the reader.

*L64 – please list the observational sources which are the only estimate available*
Only the Zemp et al. studies provide estimates for the ice sheet peripheral glaciers – we have added this in the text to make this clearer.

*L67 – How is the uncertainty propagated in time? This is a more general question as it is general hard to grasp from the MS how uncertainty is propagated, both for the individual estimates and the total imbalance. Please add information in the relevant sections of the manuscript*

We accumulate the uncertainty over time as the root sum square of the annual uncertainties – we do mention this in this sentence but we appreciate it might not be clear, so we have amended our wording slightly.

As we mentioned previously, we have also expanded on the computation of our uncertainty estimates, where requested in your specific comments, in the hope that this is now more clear.

*Fig2 – please change the colors so the studies can be separated*

We have modified the colours used and increased the line thickness on Figure 2b so they are more easily separated.

*L72 – What is a close agreement? Please report the magnitude of 1 standard deviation, it is not readable from the figure. Maybe add a table (maybe in supp.)*

Thank you for your suggestion – our preference is not to create a supplementary section and to have all the information in the main text so it is readily available. With this in mind we have added the standard deviation in all regions included in our inter-comparison in the text.

*L77 – 10 ,130pm1713 Gt -> -10,130pm1,713 or -10130pm1713. This is a general comment and should follow the same convention throughout the MS. You also give Tt in some cases please consider the number of significant digits.*

Done.

*L96 – Much of the uncertainty in the ice thickness is in the areas where it is modeled. "measured"-> "estimated".*

Done.

*L117 – this statement with the reference to fig 3 is hard to follow as figure 3 only shows the peninsula. Please be specific as to which ice shelves are included in the study.*

We have amended the text to clarify that the ice shelves in question are at the Antarctic Peninsula. We have redrawn Figure 3 which more clearly labels the ice sheets included.

*Figure 3 – the color bar for time is not readable*

We have redrawn this figure so it is more readable

*L120 – Again, how is figure 3 informing on calving? Please add information in the caption to help the reader to better understand the figure*

We have amended the caption to explain that "Iceberg calving is calculated as the difference in area between successive barrier positions."

*L127 – A polynomial-fit to elevation data is not trivial, which parameters are included in this, please elaborate on this equation and give references*

We have rewritten the text to clarify that the ice shelf thickness change data were sourced from Adusumilli et al. (2020) and computed following the method of Paolo et al., (2015). We have summarised their method, and added a note to say that "Full details of the methods used in this calculation can be found in Paolo et al. (2015)."

*L130 – The altimetry is adjusted for firn air, guess this is from RACMO, but not mentioned in the text. Additional information about the firn would be of high relevance.*

As the data and method are published, we do include citations to support each stage of the processing. We do however summarise the approach and refer the reader to Paolo et a., (2015) for further details, where the firn model is described in full.

*L150 – The reference guidelines https://www.the-cryosphere.net/submission.html#references states "Informal or so-called 'grey' literature may only be referred to if there is no alternative from the formal literature". Please give an argument for not having other references and therefore need to use "personal comm". This appears multiple places in the MS. Then I will leave it with the editor to see if this is justified.*

There are two instances of "personal comm" which corroborate the PIOMAS and GIOMAS ice density being equal to 917 kg/m³. Axel Schweiger and Jinlun Zhang both confirmed via email that 917kg/m³ was the fixed ice density used in PIOMAS and GIOMAS, however we could not find this value in any publications. We double-checked this with Schweiger and Zhang, asking whether any such publication exists, and they confirmed that there are none. We are therefore unable to reference the 917 kg/m³ value for ice density without grey literature.

The final instance of "personal comm" followed the statement "We assumed an error on monthly PIOMAS/GIOMAS volume of 1,800 km³" This sentence has now been expanded in order to also address comment 2.25, and no longer includes the grey reference.

*The sea ice section more generally: For the density of sea-ice "glacier-ice" density is chosen for PIOMAS and GIOMAS, which is normally associated with first-year ice. Timco and Frederking (1996) report "in situ density of first-year sea ice range from 0.84 to 0.91 Mg m−3 for the ice above the waterline, and 0.90 to 0.94 Mg m−3 for the ice below the waterline." Please comment on the volume conversion and the effects on the uncertainty conversion. The neglect of multi-year ice densities in the model estimates (multi-year is included in Tilling's CS2 estimate) seems to overestimate the mass derived from the models. This should at least be included in the uncertainty.*

The value of 917kg/m³ was chosen because this is the (fixed) density used in the PIOMAS / GIOMAS models to attribute a volume to the simulated ice growth. Schweiger et al. (2011) estimate the monthly error on PIOMAS volume by a range of methods including comparison to in-situ. The error they derive will therefore inherently account for errors in ice density (if the wrong ice density was used then the volume will not compare well with the in-situ data). Therefore using a different ice density other than that programmed into the model to convert from volume to mass would be inappropriate. We have amended the text so that this is made clear to the reader.

*L165 – how is this uncertainty derived?*

The errors on PIOMAS volume for October and March are 1,350 and 2,250 km³, estimated in Schweiger et al., (2011) using a range of methods including comparison to in-situ data and model sensitivity analyses. We take the average of these (1,800 km³) as the uncertainty for all months, and in the absence of a formal uncertainty budget for GIOMAS we assign it the same monthly uncertainty, which seems a reasonable inference given that the PIOMAS uncertainties are themselves uncertain (Schweiger et al., 2011). We have added this explanation to the text.

*L186 – The text states 1.2pm0.3 and the table states 1.2pm0.9, which is the right number?*

Thanks for spotting this, the text is a typo: the correct number is in the table (please note the table is 1196 + 90 Gt/yr which is not 1.2 ± 0.9 Tt/yr). We have fixed this.

*Table 1 – As the periods are not the same it would be informative if the numbers could be given both as rates and totals in this table.*

Thank you for this suggestion - however we feel adding totals would almost double the amount of information in this table (which is already large), mix mass rates and cumulative mass change, and make it unwieldy and potentially confusing for the reader. The total mass change for each component is already presented separately to the reader, both in the text and in Figure 4, so we feel it is not necessary to add here.

*L220 – Is the temperature of -20 used for all ice bodies? E.g. -20 seem cold for sea-ice with snow on top and water below*

The energy required to melt Earth's ice is primarily associated with phase transition (333,000 J/kg), with only a small contribution due to warming to the melting point (2108 J/kg °C). The fractional energy required for warming is therefore 0.68 % per °C. For ice at -30, -20 and -10 °C, the fractional energy is 19 %, 12 % and 6 %, respectively. In the absence of reliable temperature data, we choose -20 °C as a mean value for ice shelves and sea ice. However, you're correct to raise this as an uncertainty, so we have included an uncertainty of 10 °C in our energy budget calculation. We have amended the text to describe the fractional energy sensitivity and the temperature uncertainty.

*L222 – What is the uncertainty on this number?*

As per our response to your previous comment, we have added uncertainties, now including an estimated uncertainty in temperature.

*Figure 4 – Figure 4 please add the cumulative uncertainty.*

We have added the mass change and estimated cumulative uncertainty for each individual ice component, and their sum, at the end of the study period as shaded bars.

[revised manuscript text omitted]

**Figure 2 (a) Cumulative mass change for glacier regions (Gt) between 1962 and 2019. Outlines of the glacier regions (RGI 6.0) are marked by black lines and glacierized areas are indicated in orange: ACN, Arctic Canada North (area 105,110 km²); ACS, Arctic Canada South (40,888 km²); ALA, Alaska (86,725 km²); ANT, Antarctic and Subantarctic (132,867 km²); CAU, Caucasus and Middle East (1,307 km²); CEU, Central Europe (2,092 km²); GRL, Greenland (89,717 km²); HMA, High Mountain Asia (97,606 km²); ISL, Iceland (11,059 km²); NZL, New Zealand (1,161 km²); RUA, Russian Arctic (51,591 km²); SAN, Southern Andes (29429 km²); SCA, Scandinavia (2,949 km²); SJM, Svalbard and Jan Mayen (33,958 km²); TRP, Low Latitudes (2,341 km²); WNA, Western Canada and USA (14,524 km²). (b) Glacier rate of mass change (Gt yr$^{-1}$) in regions where estimates from different techniques are available, including satellite altimetry (Foresta et al., 2016; Jakob et al., 2020; Tepes et al., 2020), extrapolation of in-situ glaciological and geodetic data (Zemp et al., 2019, 2020), satellite gravimetry (Wouters et al., 2019),  satellite InSAR (Braun et al., 2019), and satellite stereo imagery (Dussaillant et al., 2019; Shean et al., 2020) over the period 2010-2015. The reconciled estimate (calculated as the average of the estimates available in a given region and year) is shown in grey.**

615   620   625

[Figure]

**Figure 3 Change in Antarctic ice shelf barrier position (left) and thickness (right) over time. Barrier positions are derived from episodic satellite imagery (Cook and Vaughan, 2010), and barrier thicknesses are derived from airborne ice penetrating radar (light grey lines) and satellite radar altimetry (Fretwell et al., 2013). Iceberg calving is calculated as the difference in area between successive barrier positions.**

630

[Figure]

**Figure 4 Global ice mass change between 1994 and 2017 partitioned into the different floating (blues) and grounded (purples) components. Shaded bars indicate the cumulative mass change and estimated uncertainty for each individual ice component (blues, purples) and their sum (black). The equivalent sea-level contribution due to the loss of grounded ice from Antarctica, Greenland and mountain glaciers is shown in the y-axis on the right hand side.**

635

---

## Author Comment (AC2) · 15 Nov 2020

Dear Roger,

Thanks for your comment on our manuscript. We estimate that the total rate of ice melting has increased in each of the three 5-year intervals for which we have complete sampling (Table 1). Unfortunately, the record is too short at present to establish whether the rate of increase is linear or otherwise.

Kind regards, Tom Slater, on behalf of the authors.
* * *